# Changing role of coral reef marine reserves in a warming climate

Nicholas A. J. Graham [1✉], James P. W. Robinson[1], Sarah E. Smith[1], Rodney Govinden [2], Gilberte Gendron[3,4] & Shaun K. Wilson[5,6]

Coral reef ecosystems are among the first to fundamentally change in structure due to climate change, which leads to questioning of whether decades of knowledge regarding reef management is still applicable. Here we assess ecological responses to no-take marine reserves over two decades, spanning a major climate-driven coral bleaching event. Pre-bleaching reserve responses were consistent with a large literature, with higher coral cover, more species of fish, and greater fish biomass, particularly of upper trophic levels. However, in the 16 years following coral mortality, reserve effects were absent for the reef benthos, and greatly diminished for fish species richness. Positive fish biomass effects persisted, but the groups of fish benefiting from marine reserves profoundly changed, with low trophic level herbivores dominating the responses. These findings highlight that while marine reserves still have important roles on coral reefs in the face of climate change, the species and functional groups they benefit will be substantially altered.

[1] Lancaster Environment Centre, Lancaster University, Lancaster LA1 4YQ, UK. [2] Seychelles Fishing Authority, Victoria, Mahe, Seychelles. [3] Seychelles National Parks Authority, Victoria, Mahe, Seychelles. [4] Sustainable Ocean Seychelles, Beau Belle, Mahe, Seychelles. [5] Department of Biodiversity, Conservation and Attractions, Perth, WA 6151, Australia. [6] Oceans Institute, University of Western Australia, Crawley, WA 6009, Australia. ✉email: nick.graham@lancaster.ac.uk

Climate change is altering the composition and structure of coral reef ecosystems[1,2], with repeated coral bleaching events leading to widespread loss of live coral cover[3]. Following these climate-driven disturbances, some reefs have undergone regime shifts to alternate dominant taxa such as seaweeds, while others have become characterised by stress tolerant and rapidly recovering coral species[4,5]. Similarly, coral reef fishes respond to benthic changes in differing ways, dictated by ecological and life history traits[6]. Collectively, these non-random species responses are leading to novel ecosystem configurations, and a recognition that coral reef communities are changing substantially in the Anthropocene and are unlikely to return to the stable compositions experienced throughout the Pleistocene[1,2,7]. A fundamental question, which we currently have little understanding of, is how effective current conservation and management approaches will be in these novel ecosystem contexts[1,7].

No-take marine reserves are one of the most pervasive and successful conservation and management tools used in coral reef ecosystems, where many species are sedentary or territorial, and thus respond well to spatial management[8]. Evidence from the past 30 years of research on no-take marine reserves, identifies a range of beneficial ecological responses[9,10]. Benthic cover of live coral is often higher inside reserves, particularly in the absence of major disturbance[11,12]. However, the greatest responses are for reef fish, which are protected from fishing. Species richness of the fish community is typically greater inside marine reserves[9], building up over relatively short timescales[13]. The largest differences in the fish community are detected for fish biomass, which based on a global meta-analysis can be >400% greater in reserves compared to fished areas[9], with this biomass enhancement driven primarily by higher trophic level carnivorous fishes[14,15]. Whether these expectations of marine reserves, and the knowledge we have built up about them, is now changing is a key question, especially as marine reserves in tropical oceans are expected to experience sea surface temperatures outside historical ranges by mid-century[16].

Here we use a unique opportunity to understand the role of marine reserves under changing environmental conditions on coral reefs. Seychelles was severely impacted by the 1998 global coral bleaching event, when >90% loss of live coral caused major short term impacts on benthic structure and fish diversity[17], and longer-term divergent trajectories where some reefs underwent regime shifts to macroalgal dominance while others recovered to hard coral dominance[5]. We assess the changing role of no-take marine reserves in Seychelles from before to 16 years after this major climatic-disturbance event. Our data span 21 reefs across the entire range of the populated inner Seychelles Islands, including nine reefs within no-take marine reserves (Supplementary Fig. 1; Methods). These marine reserves are some of the oldest globally, established between 1968 and 1979. Their effectiveness in promoting greater fish biomass was established in the 1990's prior to the 1998 coral bleaching event[18]. Specifically we ask: (1) Does the coral reef benthic response to marine reserves change following major coral bleaching?; (2) How does major coral bleaching influence the role of marine reserves in enhancing fish species richness?; and (3) Are reef fish biomass and trophic structure maintained in marine reserves following coral bleaching?

We find that prior to coral bleaching, the ecological effects of the marine reserves were consistent with expectations, with greater coral cover, higher species richness, and enhanced fish biomass driven in part by higher trophic level fishes. However, following coral bleaching, there was no difference in coral or macroalgal cover between fished reefs and reefs in marine reserves. Species richness effects of marine reserves were

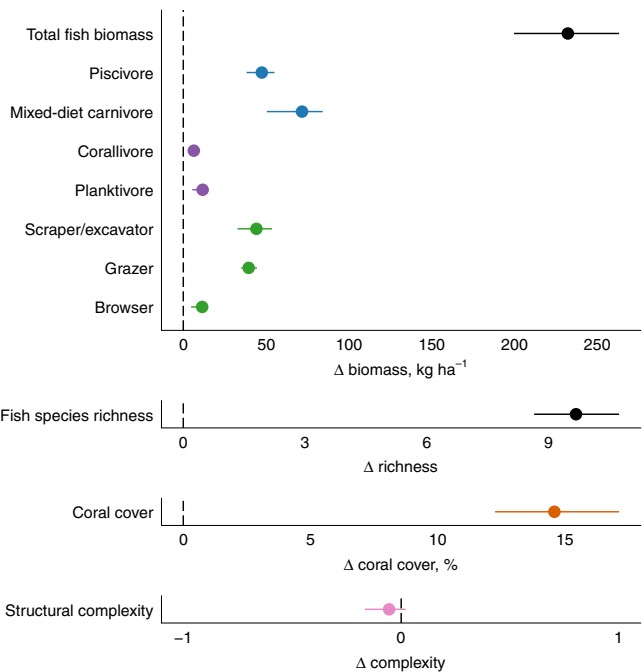

**Fig. 1 Marine reserve effects before mass coral bleaching.** Points are the average change in each ecological variable on nine reef reserves relative to 12 fished reefs (0 = equal values on reserve and fished reefs). Error bars are drop-one jackknife confidence intervals showing the variance in reserve effects attributable to individual reef sites. Triangles denote 'change in'. Pre-bleaching macroalgal cover was negligible at all 21 sites and is excluded here (mean = 1.2% ± 5.66 standard error). Corresponding log response ratio values provided in Supplementary Fig. 2. Source data are provided as a Source Data file.

diminished, and declined through time for reefs where coral cover recovered. Greater fish biomass persisted following coral bleaching, but the feeding groups of fish benefiting substantially changed. Upper trophic level piscivores and to a lesser extend mixed-diet carnivores showed little or much reduced reserve effects, with the enhanced biomass in marine reserves driven predominantly by lower trophic level herbivores. Our findings highlight that while marine reserves still drive ecological responses on climate-impacted coral reefs, the groups of species that benefit are substantially different from expectations.

## Results and discussion
**Pre-bleaching marine reserve effects.** In 1994, prior to the major coral bleaching event of 1998, Seychelles' marine reserves were performing similarly to many others globally. Fish biomass was on average 75% greater in marine reserves than fished areas (Fig. 1, Supplementary Fig. 2). This greater biomass was predominantly driven by higher trophic level piscivores and mixed-diet carnivores, including species from the grouper and snapper families. Scraper/excavator and grazing herbivore species also had greater biomass in marine reserves, as did planktivores, corallivores and macroalgal browsers to lesser extents (Fig. 1, Supplementary Fig. 2). Greater fish biomass, and a strong response for higher trophic level carnivores has been an established effect of marine reserves across the Indo-Pacific[14,19] and the Caribbean[15,20]. The reefs in marine reserves also had a higher density of species (10 more species per replicate; 154 m$^{-2}$) than fished reefs, again a well-established response of fish communities to protection[9,10,13]. Coral cover was on average 15% greater in marine reserves, while macroalgal cover and the structural complexity of reefs did not differ between fished and protected reefs.

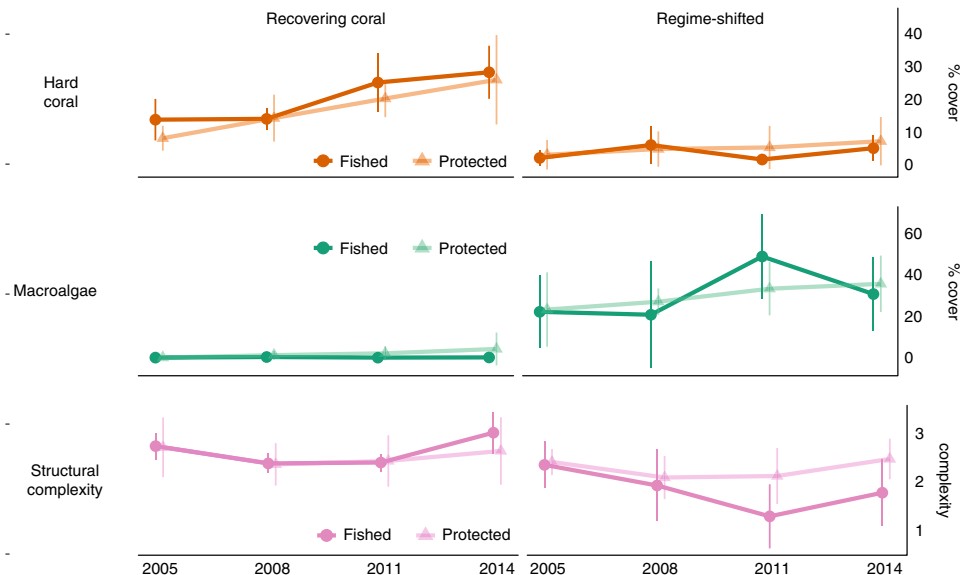

**Fig. 2 Change in benthic habitat composition after coral bleaching.** Points are mean values (±2 standard error) for percent cover of hard coral (orange) and macroalgae (green), and habitat structural complexity (pink) over 2005–2014, for recovering coral (left panel, $n = 12$) and regime-shifted reefs (right panel, $n = 9$). Fished reefs are dark-shaded and reserve reefs are light-shaded. Note different $y$-axis ranges for different benthic taxa. Source data are provided as a Source Data file.

Greater coral cover in protected reefs prior to coral bleaching may be due to placement of the reserves in areas of high cover for tourism opportunities, protection from the impacts of destructive fishing gears, or faster recovery rates from disturbances[11,12,21].

**Post-bleaching benthic responses**. Following the 1998 bleaching event, Seychelles' reefs underwent one of two well-documented trajectories. While 12 of the 21 reefs surveyed recovered to hard coral-dominance, the other nine underwent regime shifts to fleshy macroalgae[5,22]. These post-bleaching trajectories are apparent on both fished and protected reefs, with no difference in cover of coral or macroalgae between marine reserves and fished reefs in the years post-bleaching (Fig. 2). Indeed, marine reserve status was a poor predictor of post-bleaching reef trajectory and coral recovery rates in Seychelles[5]. While the structural complexity of reefs was greater on recovering compared to regime-shifted reefs, again this benthic variable did not differ between fished or protected reefs.

**Post-bleaching fish biomass and richness responses**. Reserve effects on fish biomass following bleaching were similar to pre-bleaching levels on macroalgae-dominated reefs, whereas on recovering reefs the reserve effect diminished 10 years post bleaching, before showing some increase (Fig. 3a). These relative effects are partially due to the differing rates at which biomass re-accumulated on reefs post-bleaching. In recovering habitats, biomass increased faster on fished reefs (79% increase from pre-bleaching levels by 2014, 7% year$^{-1}$ with 95% certainty interval [2.3, 10.4]) than on protected reefs (52% increase by 2014, 4% year$^{-1}$ [−0.4, 8.5]), which diminished the reserve effect (Fig. 3c, Supplementary Fig. 3a). Biomass was less impacted by bleaching on macroalgal-dominated reefs, remaining at pre-bleaching levels in marine reserves and increasing by 38% in fished areas by 2014, but with uncertain temporal trajectories that were close to zero (fished = 2% year$^{-1}$ [−1.9, 6.5]; protected = 0% year$^{-1}$ [−4.4, 4.6]) (Fig. 3c, Supplementary Fig. 3b). The increase in fish biomass, seen most strongly on fished recovering reefs, was driven primarily by scraping herbivores (parrotfishes), and to a lesser extent mixed-diet carnivores (Supplementary Fig. 4). Parrotfishes

have enhanced growth rates following coral mortality events[23], likely due to the expansion of nutritional resources driven by succession of microbial photoautotrophs on or in the carbonate substrate[24]. This helps explain the common phenomenon of increased herbivore biomass following disturbance[6]. Similarly, dietary resources for mixed-diet carnivores, which target a diverse array of invertebrate prey[25], may increase on rubble-dominated and degraded reefs, where macroalgae does not dominate space[26]. Conversely, fishing pressure did not decrease over this same time period[27] and is unlikely to have influenced changes in fish biomass.

The effectiveness of marine reserves in enhancing fish species richness was diminished on post-bleaching reefs (Fig. 3b). While marine reserves maintained higher fish species richness on regime-shifted reefs in most years, these reefs remained depauperate compared to pre-bleaching levels throughout the post-bleaching recovery period (Supplementary Fig. 3d), with richness recovery rates close to zero (fished = 0.9% year$^{-1}$ [−0.9, 2.7]; protected = −0.2% year$^{-1}$ [−2.0, 1.7]). In contrast, the net reserve benefit on recovering reefs gradually declined to no difference by 2014, which was driven by slightly faster accumulation of species on fished reefs (3% year$^{-1}$ [1.7, 4.9]) than protected reefs (2% year$^{-1}$ [0.4, 3.9]) to match protected richness levels by 2014 (Fig. 3d, Supplementary Fig. 3d). The recovering reefs were characterised by increases in keystone structure provided by branching corals, which favours survivorship and increases in small-sized fishes and small-size classes of larger species[28]. Further, the increases in biomass of scraping herbivores and mixed-diet carnivores described above, also resulted in increases in species richness for these groups[22]. In sum, while the expectation[9] of reserves enhancing fish biomass was generally maintained post-bleaching, their role in enhancing species richness relative to fished reefs was only retained on macroalgal-dominated reefs.

**Re-organisation of fish trophic structure in marine reserves**. The sustained effect of marine reserves on biomass of fish communities belies a substantial re-organisation of the trophic groups of fishes benefiting from protection. The dominant pre-bleaching

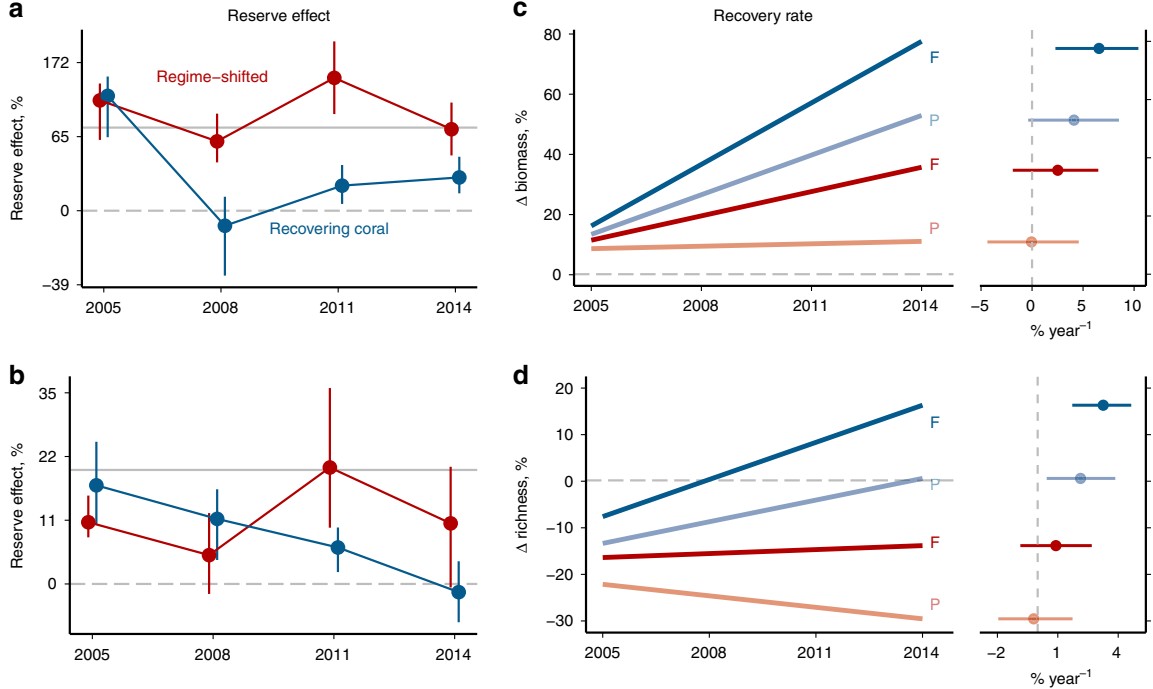

**Fig. 3 Post-bleaching marine reserve effects on fish biomass and species richness from 2005–2014.** Reserve response ratio (reserve/fished) for biomass (**a**) and richness (**b**) on a percentage scale (0 = no effect), with jackknife resampling error bars and grey lines indicating no reserve effect (dashed) and pre-bleaching reserve effect (solid). Recovery of biomass (**c**) and richness (**d**) relative to pre-bleaching levels (y = 0, dashed grey line), for fished (F) and protected (P) reefs. Lines are posterior median predictions for each reef regime and reserve status, with points showing the posterior median rate of biomass or richness recovery per year (±95% certainty intervals) (n sites = 7 fished-recovering, 5 protected-recovering, 5 fished-shifted, 4 protected-shifted). See Supplementary Fig. 3 for corresponding observed biomass and richness values, and Supplementary Figs. 5–6 for biomass and richness recovery trajectories at each reef. Source data are provided as a Source Data file.

biomass effects for piscivores (Fig. 4a) gave way to scraper and excavator fish, whilst effects for mixed-diet carnivores remained positive on protected recovering reefs post-bleaching (Fig. 4b). Scrapers and excavators, which make up the majority of the parrotfish family and contribute to artisanal fisheries outside reserves in Seychelles[27], have been shown to increase in abundance following coral mortality[6], likely explaining their higher biomass in reserves post-bleaching. Scrapers and excavators provide important processes on coral reefs such as grazing of algae and the bioerosion of dead reef framework[29]; processes that are likely enhanced within these reserves. On regime-shifted reefs, the biomass effect size between marine reserves and fished reefs was largest, but in this case macroalgal browsing species, such as species of rabbitfish (Siganidae), benefitted the most from protection (Fig. 4c). Other herbivorous fish species (scrapers, excavators and grazers) also contributed substantially to the reserve biomass effect on regime-shifted reefs post-bleaching. Enhanced algal resources[30], together with some herbivore species having very fast growth and maturation rates[31], most likely contributed to these unexpected marine reserve fish benefits. In contrast, piscivores, which are so often an important component of fish communities in marine reserves pre-coral bleaching[14], did not benefit from reserves post-bleaching on either recovering or regime-shifted reefs (Fig. 4b, c). This is quite likely due to population declines driven by erosion of their prey base following reef structural loss[32,33], coupled with slow life histories of species which require long recovery times in intact coral habitats[34].

To investigate the relative role of marine reserves in enhancing carnivorous versus herbivorous processes, we grouped species into those that feed on fish or invertebrates, and those that feed on algal resources. Pre-bleaching, the marine reserve effect was

similar for both groups, with biomass 62% greater for herbivores and 77% for carnivores, and both groups also reached similar biomass levels in protected areas (carnivore = 259 kg ha⁻¹ [194, 347]; herbivore = 266 kg ha⁻¹ [198, 347]) (Fig. 5a). While marine reserves still had a weak positive influence on the biomass of carnivorous fishes post-bleaching, driven primarily by mixed-diet carnivores (Fig. 4b, c, Supplementary Fig. 4), there was substantially greater herbivore biomass than in pre-bleaching marine reserve conditions for both recovering and regime-shifted reefs (Fig. 5b, c). Specifically, in marine reserves, herbivore biomass reached 513 kg ha⁻¹ [317, 793] on recovering reefs and 472 kg ha⁻¹ [349, 606] on regime-shifted reefs, substantially exceeding carnivore biomass (recovering = 309 kg ha⁻¹ [192, 505]; shifted = 243 kg ha⁻¹ [170, 312]). Gains in herbivorous reef fish following coral mortality have been documented before[6,22], and here we demonstrate the role marine reserves play in greatly enhancing these fish groups in novel post-disturbance coral reef configurations[2].

**Uncertain future for coral reef management.** Marine reserves have been a mainstay of the coral reef conservation toolbox for several decades. Here we demonstrate that climate change impacts are fundamentally altering the composition of coral reefs, and the benefits marine reserves can offer. Any influence of protection on the reef benthos was absent post-bleaching, possibly swamped by the overwhelming impact of climatic disturbance[35], or because the processes of coral recruitment and growth that drive benthic recovery[36] are independent of protection from fishing. There is uncertainty regarding what drove higher coral cover in marine reserves pre-bleaching, and reserve

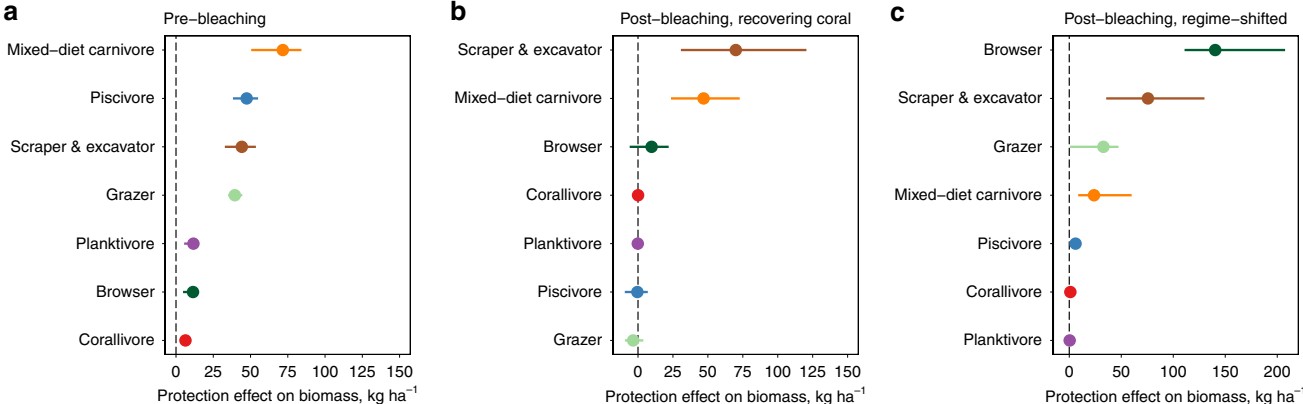

**Fig. 4 Marine reserve effects on fish functional groups pre- and post- bleaching.** Points are the mean difference in biomass between protected and fished areas, with drop-one jackknife error bars representing the uncertainty in reserve effects attributable to individual reef sites. **a** Higher fish biomass in marine reserves pre-bleaching was primarily driven by mixed-diet carnivores, and piscivores, with smaller contributions from scrapers & excavators and grazers. All groups had higher biomass in the 9 reserves compared to 12 fished reefs. **b** On the 12 reefs that recovered coral cover following major bleaching, higher biomass in reserves compared to fished areas was now dominated by scraper and excavator fish, followed by mixed-diet carnivores. Several functional groups show no reserve effect. **c** On the nine reefs that underwent regime shifts to macroalgae, the greatest biomass difference between fished and protected reefs was for macroalgal browsing fish, followed by scraper & excavator and grazer groups. Source data are provided as a Source Data file.

placement in higher coral cover locations could have been a factor. However, for one of the marine reserves, which represented three of our studied reef sites, protection was driven by bird conservation on the island. High coral cover habitat was therefore not a factor in reserve selection here, although, even at these sites, coral cover pre-bleaching was ~15% greater than the average on fished reefs.

The role of marine reserves in enhancing fish biodiversity, captured here simply as species richness, appears to be greatly diminished. While marine reserves can increase species richness in heavily fished seascapes[13], the diversity of fish on many coral reefs is increasingly governed by disturbance-mediated changes to habitat[37]. Perhaps most significantly, while the important role of marine reserves in enhancing fish biomass continued post-bleaching, the groups of fish benefiting from the protection drastically changed. A major shift from carnivorous to herbivorous reef fish benefitting from marine reserve protection implies shifts in trophodynamics and ecosystem services. Herbivores deliver key ecosystem functions[29], and are important components of fisheries in many geographies. Indeed, herbivorous fish have sustained reef-associated trap fisheries landings in Seychelles following the 1998 coral bleaching event[27]. The role of marine reserves in exporting herbivorous fish to fisheries through adult spillover or larval export is likely to be substantial, and could play a key role in continued food security as coral reefs degrade through climatic impacts[38].

The success of new marine reserves will likely benefit from consideration of individual reef exposure and response to climate change and other increasing pressures characterising the Anthropocene. For example, marine reserve placement may be most effective in locations where the rate of warming and threat of frequent bleaching is lowest[39], where coral reefs are likely to remain most 'functional'[40], where recovery from climate disturbance is most likely[5], or where reef conditions are outperforming expectations given local drivers such as proximity to markets[41]. Of note, emerging benthic configurations such as those dominated by macroalgae may still provide important diversity and fishery benefits[27,42] and, as we show here, also respond positively to marine reserve management. These results may be particularly relevant to Caribbean coral reefs, where shifts from coral to macroalgae have been quite widespread[43].

The changing ecological responses to marine reserves that we document are not likely to be short-lived, given the time between severe coral bleaching events is shortening, and coral reefs are increasingly existing in boom-bust live coral cycles or shifting to alternative habitat-forming taxa such as macroalgae[3,28]. As the scale and frequency of threats to coral reefs increase, so too must the practices and institutions that aim to sustain them[44,45]. While marine reserves will be part of this solution, urgent reductions in global greenhouse gas emissions and mitigation of other underlying social and institutional pressures on reefs will be critical[1,46]. Unexpected ecological compositions on coral reefs are also changing the ecosystem services relied upon by millions of people across the tropics[47]. To be successful and equitable in such novel environments, reef conservation and management must strive for continued provision of ecosystem services, livelihoods and food security[47,48]. Our findings indicate that ecological responses to other common management approaches on coral reefs and other climate-sensitive ecosystems may also need to be carefully re-evaluated.

## Methods
**Study location and data collection**. Using a 20-year data set (1994–2014), spanning a major climate-induced bleaching event in 1998, we assess the long-term ecosystem dynamics of 21 reef sites across the inner islands of Seychelles. Seychelles' reefs were among the most severely affected globally by the 1998 coral bleaching event, in which a strong El Niño coincided with a strong positive phase of the Indian Ocean Dipole[49]. Nine sites were located within four no-take marine reserves, which were established between 1968 and 1979; Cousin Special Reserve, Ste Anne Marine Park, Baie Ternay Marine Park, and Curieuse Marine National Parks. Enforcement of these marine reserves is high, and has been consistent since their establishment; two of the reserves have resident rangers on the island they encompass, while the other two are patrolled daily by the Seychelles National Parks Authority. The remaining 12 sites were located in fished areas. Fishing fleet size has remained stable, or increased for one gear and island, through the time period of this study[27]. The major changes to the reefs have been associated with benthic changes precipitated by the 1998 bleaching event[5].

Sites were surveyed using identical methods in 1994, before the mass coral bleaching of 1998, and in 2005, 2008, 2011 and 2014, following the bleaching event. Eight to sixteen replicate 7 m radius point counts were surveyed along the reef slope on each reef, covering up to 0.5 km of reef front and 2500 m$^2$ of reef habitat each survey year. Fish abundance and individual body length (to the nearest cm) of diurnally active non-cryptic species (134 species from 16 families) were estimated in each point count area using underwater visual census (UVC). Larger, mobile species were recorded before smaller, more site-attached species to minimize the likelihood of double counting individuals that left and re-entered the survey area.

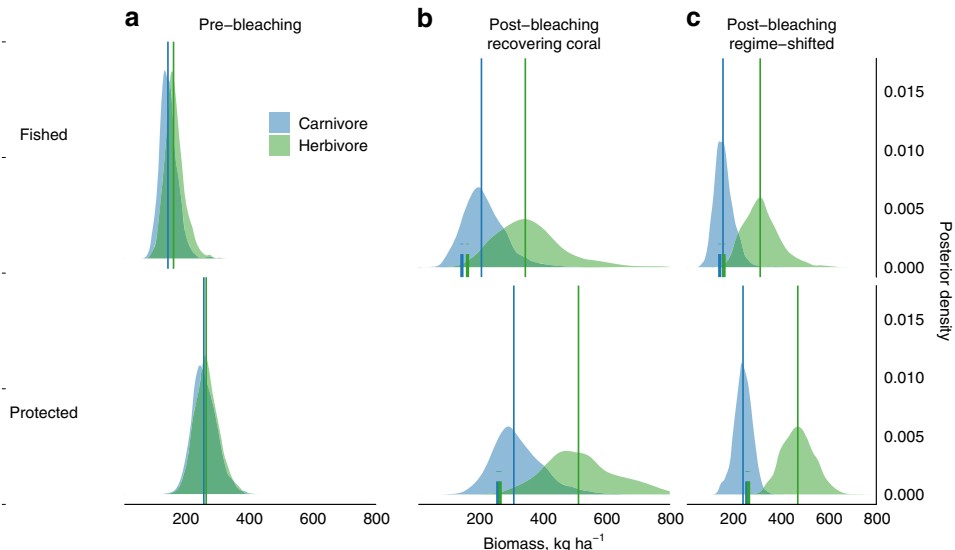

**Fig. 5 Marine reserve effects on carnivore and herbivore biomass before and after coral bleaching.** For fished (top) and protected (bottom) reefs, density curves are the posterior distributions of carnivore (blue) and herbivore (green) biomass before (**a**) and after bleaching for recovering coral (**b**) and regime-shifted (**c**) reefs. Vertical lines indicate median biomass, with short lines indicating median 1994 biomass on post-bleaching (**b**, **c**). See Supplementary Figs. 7–9 for pre- and post-bleaching posterior distributions at each reef, Supplementary Fig. 10 for posterior distributions including 2017 survey data, and Supplementary Fig. 11 for observed biomass trends.

Survey time was not held constant, and varied according to the abundance and diversity of fish in each replicate. The accuracy of fish body length estimations were assessed at the start of each sampling day in all sample years. The length of random sections of PVC pipe were visually estimated prior to the start of data collection, and compared to the actual length of the pipe. Length estimates were consistently within 4% of actual lengths[50]. We converted data on fish counts to biomass with published length–weight relationships[51,52], and fish were placed into trophic groups based on feeding preferences[32].

Within each point count area, the percent cover of live hard coral, soft coral, macroalgae, sand, rubble, and rock was quantified visually and using 10 m line intercept transects[53]. The structural complexity of the reef was visually estimated on a 6-point scale[15]. This structural complexity measure captures landscape complexity, including that provided by live corals, the underlying reef matrix, and other geological features, and has been shown to correlate well to other measures of complexity, such as reef height and the linear versus contour chain method[53].

A major global bleaching event in 2016 severely impacted Seychelles' reefs[5]. While data were also collected in 2017, a number of sites could not be visited, including three of the four regime-shifted marine reserve reefs. We have therefore included data in our analyses up to 2014 to (1) ensure our study includes reef sites which were consistently monitored in the post-bleaching period, and (2) avoid skewing analyses towards recovering reef characteristics. However, while recognizing the smaller sample size in 2017, the data indicate that the post-1998 bleaching trends do continue to 2017 (Supplementary Figs. 4, 10, 11). Thus, our data indicate that the altered role of the marine reserves were established following the first major bleaching disturbance of 1998, and may persist through subsequent climate disturbances.

**Coral reef status.** A regime-shifting reef was defined as one where post-disturbance macroalgal cover becomes greater than coral cover and trajectories through time indicate that cover of macroalgae remains high or is increasing. Recovering reefs, conversely, are defined as those reefs where post-disturbance coral cover becomes greater than macroalgal cover and remains greater or increases through time[5]. Among fished sites, seven were classified as recovering and five as regime-shifting. Among protected sites, five were classified as recovering and four as regime-shifting.

**Marine reserve effects.** We examined marine reserve effects on community biomass, functional group biomass (kg ha$^{-1}$), and species richness of coral reef fishes. For each variable, we measured (1) the average difference between fished and protected reefs and (2) the reserve response ratio (protected/fished). We used a drop-one jackknife approach to estimate uncertainty in reserve effects. Specifically, we estimated the difference between fished and protected reefs after excluding one site, and repeated the process by excluding each site in turn (i.e., 21 jackknife estimates per survey year). The minimum and maximum jackknife estimates represent uncertainty in reserve effect estimates introduced by individual reefs with exceptionally high or low values for fish biomass or species richness. For benthic

changes, we estimated the mean benthic cover (% hard coral, % macroalgae) and structural complexity in each year, separately for fished and protected reefs in recovering coral and regime-shifted habitats.

**Modelling reef-level recovery trajectories.** We modelled post-bleaching recovery trajectories in fish biomass and species richness at the scale of individual reefs. For each reef, we converted fish biomass and species richness into the absolute difference between pre- and post-bleaching values (i.e. 1994–post-bleaching values in 2005–2014). Using these response data for $y$, we fitted four linear models with structure:

$$y_i \sim \mathrm{N}(\mu_i, \sigma) \tag{1}$$

$$\mu_i = \beta_{0,r} + \beta_1 \mathrm{habitat}_i + \beta_2 \mathrm{year}_{i,r} + \beta_3 \mathrm{management}_i + \beta_4 \mathrm{habitat} * \mathrm{year}_i + \beta_5 \mathrm{management} * \mathrm{year}_i \tag{2}$$

for each survey $i$ conducted from 2005 to 2014. Fixed covariates were habitat (recovering or regime-shifted), survey year (2005–2014, mean centered), and fishing management (protected or fished). We accounted for temporal correlations within reefs by fitting separate survey year slopes and intercepts for each reef $r$, and fitted interacting fixed effects between survey year*habitat, and survey year*reserve status to estimate linear trends in each habitat regime for fished and protected areas. Priors were weakly informative (fixed covariates = N(0, 10); σ = Cauchy(0, 2)). Predictions were visualized by drawing 1000 samples from the posterior distribution across each survey year, for each fishing and habitat combination, and extracting the posterior mean and 95% certainty intervals. Model fits were assessed by drawing posterior predictions for individual reefs (Supplementary Figs. 5, 6). Posterior predictions of biomass and richness absolute difference values were rescaled to represent the percent recovery from pre-bleaching levels (i.e. 0% = pre-bleaching value).

We also modelled reserve effects in promoting carnivore and herbivore biomass before and after bleaching. Using reef-level estimates of herbivore and carnivore biomass, where herbivores included all grazing, scraping, excavating and browsing species and carnivores included all mixed-diet feeders and piscivores, we assessed pre-bleaching reserve effects by fitting the linear model:

$$y_i \sim \mathrm{Gamma}(\mu_i, k) \tag{3}$$

$$\mu_i = \beta_{0,r,FG} + \beta_1 \mathrm{management}_{i,FG} \tag{4}$$

where $y$ was the carnivore or herbivore biomass (kg ha$^{-1}$) estimated in each survey $i$, and management is fished or protected effect estimated for each functional group $FG$ (carnivore or herbivore), with reef-level intercept terms ($r$). For post-bleaching reserve effects, we fitted the same model structure as (2) for Gamma-distributed $y$ (3), but including hierarchical fixed effects of habitat (recovering or regime-shifted), survey year, management for each $FG$:

$$\mu_i = \beta_{0,r,FG} + \beta_1 \text{habitat}_{i,FG} + \beta_2 \text{year}_{i,r,FG} + \beta_3 \text{management}_{i,FG}$$
$$+ \beta_4 \text{habitat} * \text{year}_{i,FG} + \beta_5 \text{management} * \text{year}_{i,FG} \quad (5)$$

These additional fixed covariates enabled us to estimate carnivore and herbivore biomass in different habitat regimes while accounting for temporal effects of survey year. Using these two models, we visualized reserve effects on carnivore and herbivore biomass by drawing from the posterior distribution of biomass under fished and protected conditions, for pre-bleaching (4), post-bleaching recovering coral (5), and post-bleaching regime-shifted reefs (5), and for each individual reef (Supplementary Figs. 7–9).

In all models, parameters were estimated by Markov chain Monte Carlo (MCMC) sampling of 7000 iterations (warmup of 1500) across three chains, using the No-U-Turn-Sampler implemented in Stan[54]. Model convergence was assessed using $\hat{R}$ values (all parameters within 0.01 of 1), the number of effective samples, and posterior predictive checks[55] (Supplementary Tables 1–2). Analyses were conducted in R 3.6.0[56], and Bayesian hierarchical models were implemented in Stan 2.18.1 using *rethinking*[55] and *RStan*[54].

**Reporting summary**. Further information on research design is available in the Nature Research Reporting Summary linked to this article.

## Data availability

We provide data associated with this study at an open source repository (https://github.com/jpwrobinson/changing-mpas). The source data underlying Figs. 1, 2, 3, 4, and Supplementary Figs. 2, 3 and 4 are also provided as a Source Data file.

## Code availability

We provide R code associated with this study at an open source repository (https://github.com/jpwrobinson/changing-mpas).

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

## Acknowledgements

This research was supported by a Royal Society University Research Fellowship awarded to N Graham (UF140691), the Australian Research Council, and the Leverhulme Trust. We thank the Seychelles Fishing Authority, Seychelles National Parks Authority, Nature Seychelles, and Global Vision International for logistical support.

## Author contributions

N.A.J.G. conceived of the study; N.A.J.G. and S.K.W. collected the data; J.P.W.R., S.E.S. and N.A.J.G. developed and implemented the analyses; N.A.J.G. led the manuscript with J.P.W.R., S.E.S., R.G., G.G., and S.K.W.

## Competing interests

G.G. is an employee of the Seychelles National Parks Authority. The other authors declare no competing interests.
