## [Peer Review File · Nature Communications]

Reviewers' comments:

Reviewer #1 (Remarks to the Author):

Overall comments:

The manuscript by Graham et al. "Changing Role of Coral Reef Marine Reserves in a Warming Climate" demonstrate that climate change impacts alter the benefits of marine reserves by showing that the group of fishes benefited by marine reserves changed. The authors also showed an absence of reserve effects for the reef benthos after major climate-driven coral mortality. Such findings are novel and have implications on the trophic ecology and ecosystem services. The data is well analyzed and the work convincing. The paper influences the field by providing a novel effect of climate disturbances on coral reef benthic and fish assemblages. The authors provide data and scripts, ensuring full transparency.

Specific comments:

Page 5: "Coral cover was on average 15% greater in marine reserves, while macroalgal cover and the structural complexity of reefs did not differ between fished and protected reefs."

What about tourism in the studied sites? Unordered tourism may cause significant impacts on benthic assemblages on coral reefs, and secondarily on fish assemblages. There is a sentence explaining that tourism influenced the establishment of reserves to higher coral cover sites.

I suggest to authors add a map with studied sites on supplementary material.

In the end of the text, the authors pointed out that new approaches should be investigated. Could the authors provide more details on at least the direction of new approaches that may be effective? This direction may be important to incentive further initiatives/studies.

METHODS

Page 17: "Enforcement of these marine reserves is high..."

Was the enforcement high since the establishment of the marine reserve? Or are there fluctuations in enforcement? Please, add more details.

Page 17: "Fish abundance and individual body length (to the nearest cm) of diurnally active non-cryptic species (134 species from 16 families) were estimated in each point count area using instantaneous underwater visual census (UVC)."

What does instantaneous means? Is there a standardized time for each sampling? If yes, how many?

Page 17: "The accuracy of fish body length estimations were assessed daily using sections of PVC pipe prior to the start of data collection, and length estimates were consistently within 4% of actual lengths³³."

Length estimated were compared over the entire period of data collection? Or was calibrated only in a fraction of samples? Please, add more details.

Page 18: "This structural complexity measure captures landscape complexity, including that provided by live corals, the underlying reef matrix, and other geological features, and has been shown to correlate well to other measures of complexity, such as reef height and the linear versus contour chain method³⁶."

I checked the reference, there is no comparison with other reef complexity measure approaches showing a correlation. Rephrase the sentence or replace the reference to a study that compared such approaches.

Figure 2: Change the symbol of one of the mean values to make the interpretation easier, mainly for those that print in B&W. Ex. a circle and a triangle.

Figure 5: Add a title to the x-axis.

Reviewer #2 (Remarks to the Author):

This paper provides an interesting chronosequence of data from inside and outside reserves, on

reefs that both recovered and remained regime-shifted following a bleaching event. The effects of marine reserves are generally clear, but it is interesting to consider their role for degraded / recovering reefs. So this paper has the potential to add to the literature.

The paper is nicely written and analysed, but my main problem is the lack of discussion of the mechanism that is underpinning the post-2005 recovery of biomass and species richness on recovering reefs (Fig S2 is critical here). Biomass is higher on protected reefs through the bleaching event, and generally continues to trend upwards as might be expected. However, the fished reefs gain biomass extremely fast after 2005, and become similar to protected reefs by 2008. This biomass gain is mirrored by an increase in species richness. What happened in 2005? Understanding this change is critical because it underpins many of the key results (eg limited effect of protection on recovering reefs). The paper demonstrates a shift in trophic structure towards herbivores, but it seems unlikely that this is the explanation given the relatively limited amount of macroalgae on these reefs and the fact that these species are part of the fishery (p6). Did fishing decrease for some reason? I think an explanation of this pattern is critical for publication in a high-profile journal in order to provide a mechanistic understanding of the changes and thus understand how applicable the results are elsewhere. [One other note: I would change the straight lines between 1994 and 2005 in Figure S2 – this seems about the least likely trajectory between these two data points given the disturbance event].

I also had an issue with the conclusion that “Any influence of protection on the reef benthos appears to have been lost,” (p8). Proving that reserves have a positive effect on reefs trajectories is challenging, requiring multiple conditions including a reserve effect on a driving factor (eg herbivore abundance) and a significant period of time between disturbances to document reef trajectories. Thus while reserve effects on coral cover are theoretically obvious, they have rarely been documented. The authors suggest reserve effects on coral cover pre-bleaching, but acknowledge this may be due to placement issues (reserves are often put on the best reefs). The results post-bleaching suggest this is more likely than an actual reserve effect since there is no difference between protected and fished reefs when all reefs are set to low coral cover levels by bleaching. Hence I don't think it is true that the influence of protection on the benthos has been lost, but rather it wasn't clear it existed at all on these reefs for whatever reason.

I think the results need to be better interpreted for what they mean for management – in particular, if someone was setting up new reserves how do these results inform siting decisions? It also seems to me worth pointing out that the regime-shifted results might be particularly relevant in the Caribbean where shifts to macroalgal dominance are typical. Furthermore, it should be pointed out that in theory the effects are only short- to medium-term – presumably, at least on recovering reefs, macroalgal cover will decrease, coral cover and structure will increase and the fish assemblage will shift back toward carnivorous species?

Given these results, I don't really see why “existing management strategies need to be re-evaluated, and the potential for new approaches should be investigated.” This may be true, but I don't see how it is supported by the results here – reserves still generally have useful, positive benefits?

Reviewer #1 (Remarks to the Author):

Overall comments:

The manuscript by Graham et al. "Changing Role of Coral Reef Marine Reserves in a Warming Climate" demonstrate that climate change impacts alter the benefits of marine reserves by showing that the group of fishes benefited by marine reserves changed. The authors also showed an absence of reserve effects for the reef benthos after major climate-driven coral mortality. Such findings are novel and have implications on the trophic ecology and ecosystem services. The data is well analyzed and the work convincing. The paper influences the field by providing a novel effect of climate disturbances on coral reef benthic and fish assemblages. The authors provide data and scripts, ensuring full transparency.

Specific comments:

Page 5: "Coral cover was on average 15% greater in marine reserves, while macroalgal cover and the structural complexity of reefs did not differ between fished and protected reefs."

What about tourism in the studied sites? Unordered tourism may cause significant impacts on benthic assemblages on coral reefs, and secondarily on fish assemblages. There is a sentence explaining that tourism influenced the establishment of reserves to higher coral cover sites.

I suggest to authors add a map with studied sites on supplementary material.

We have added a map of the study area with the study sites marked, and marine reserves delineated (Fig. S1). Tourism is focused on the marine reserves, so any deleterious impacts of tourism on the reef benthos and fish would be expected in them rather than in the fished areas, so would not explain the 15% greater coral cover in the marine reserves. Further, diving and snorkelling tourism is relatively light in Seychelles.

In the end of the text, the authors pointed out that new approaches should be investigated. Could the authors provide more details on at least the direction of new approaches that may be effective? This direction may be important to incentive further initiatives/studies.

Thank you for this suggestion. We have expanded the end of the main text by 2 paragraphs in response to this and one of the comments by Reviewer #2. We now discuss how science and reef management may need to adapt under uncertain future configurations, drawing on recent review / conceptual papers also thinking in this space (Hughes et al. 2017 Nature, Bellwood et al. 2019 Biol Conserv; Williams & Graham 2019 Func Ecol Special Issue).

METHODS

Page 17: "Enforcement of these marine reserves is high..."

Was the enforcement high since the establishment of the marine reserve? Or are there fluctuations in enforcement? Please, add more details.

Enforcement has been consistent since the start of the marine reserve establishment, with government agencies managing and patrolling three of the reserves (with resident rangers on an

island in one of them), and a local NGO having resident rangers on an island in the fourth reserve. This is now clear in the manuscript.

Page 17: “Fish abundance and individual body length (to the nearest cm) of diurnally active non-cryptic species (134 species from 16 families) were estimated in each point count area using instantaneous underwater visual census (UVC).”

What does instantaneous means? Is there a standardized time for each sampling? If yes, how many?

Instantaneous here means counting fish species in a snapshot, to avoid double counting caused by fish swimming out of and then back into the survey area. For each point count area, the larger more mobile species were surveyed first, and once recorded, any more individuals of those species that entered the area were not counted. Then the smaller more site-attached species were surveyed. As such, time per point count was not held constant, and varied according to the abundance and diversity of fish in a given survey replicate. We have now clarified this in the methods, including removing the word ‘instantaneous’, which in hindsight was not helpful.

Page 17: “The accuracy of fish body length estimations were assessed daily using sections of PVC pipe prior to the start of data collection, and length estimates were consistently within 4% of actual lengths³³.”

Length estimated were compared over the entire period of data collection? Or was calibrated only in a fraction of samples? Please, add more details.

The length estimation calibration was conducted at the start of every day’s sampling throughout the dataset (i.e. in all years). This has been clarified in the methods.

Page 18: “This structural complexity measure captures landscape complexity, including that provided by live corals, the underlying reef matrix, and other geological features, and has been shown to correlate well to other measures of complexity, such as reef height and the linear versus contour chain method³⁶.”

I checked the reference, there is no comparison with other reef complexity measure approaches showing a correlation. Rephrase the sentence or replace the reference to a study that compared such approaches.

The correlation coefficients among complexity measures in Wilson et al. (2007) Mar Biol are in Table 2. The associated results text reads: ‘Several of the methods used to estimate reef complexity were found to be strongly correlated. Visual estimates of reef topography were correlated, positively and significantly with both reef height and scores from PC1 of the holes data PCA, which were indicative of hole abundance 10–70 cm diameter (Table 2; Fig. 1). Visual estimates were also correlated significantly with rugosity, although this was a negative relationship as lower values of rugosity are indicative of greater complexity (Table 2).’

Figure 2: Change the symbol of one of the mean values to make the interpretation easier, mainly for those that print in B&W. Ex. a circle and a triangle.

As suggested we have replaced the circle with a triangle for Protected data in Figure 2. To be consistent, we have also done this for Figures S3, S4, and S11.

Figure 5: Add a title to the x-axis.

A title has now been added to the axis

Reviewer #2 (Remarks to the Author):

This paper provides an interesting chronosequence of data from inside and outside reserves, on reefs that both recovered and remained regime-shifted following a bleaching event. The effects of marine reserves are generally clear, but it is interesting to consider their role for degraded / recovering reefs. So this paper has the potential to add to the literature.

The paper is nicely written and analysed, but my main problem is the lack of discussion of the mechanism that is underpinning the post-2005 recovery of biomass and species richness on recovering reefs (Fig S2 is critical here). Biomass is higher on protected reefs through the bleaching event, and generally continues to trend upwards as might be expected. However, the fished reefs gain biomass extremely fast after 2005, and become similar to protected reefs by 2008. This biomass gain is mirrored by an increase in species richness. What happened in 2005? Understanding this change is critical because it underpins many of the key results (eg limited effect of protection on recovering reefs). The paper demonstrates a shift in trophic structure towards herbivores, but it seems unlikely that this is the explanation given the relatively limited amount of macroalgae on these reefs and the fact that these species are part of the fishery (p6). Did fishing decrease for some reason? I think an explanation of this pattern is critical for publication in a high-profile journal in order to provide a mechanistic understanding of the changes and thus understand how applicable the results are elsewhere. [One other note: I would change the straight lines between 1994 and 2005 in Figure S2 – this seems about the least likely trajectory between these two data points given the disturbance event].

Many thanks for identifying this; it has given us the opportunity to bring more of a mechanistic narrative into the manuscript. Fishing effort has not decreased during this period (Robinson et al. 2019 Nature Eco Evo). The biomass increase between 2005 and 2008 on recovering reefs was driven mainly by scraping herbivores (parrotfishes) and mixed-diet carnivores. This was shown in Fig. S9 (now S11). Recent work has identified that parrotfishes are macrophages, gaining most of their energy from microbes, and suggests that parrotfish will benefit from coral mortality (Clements & Choat 2018) A recent paper has demonstrated this, showing that parrotfish experience enhanced growth following coral mortality (Taylor et al. 2019 Glob Change Biol). Importantly, this was a pan-tropical response suggesting that these increases in parrotfish biomass seen in a number of post-disturbance reefs (Pratchett et al. 2008 OMBAR) are ubiquitous. This and the recovering coral

habitat from 2005 likely enhanced parrotfish biomass. The mixed-diet carnivores have extensive feeding opportunities on rubble-dominated and degraded reefs, where macroalgae is not dominating space (Enochs & Manzello 2012).

For the species richness changes, a lot of this increase is also due to herbivores and mixed-diet herbivores (Robinson et al. 2019 GCB). Furthermore, the recovering reefs were characterised by increasing ‘keystone structure’ provided by branching corals, which favours survivorship and increases in small-sized fishes and small-size classes of larger species (Wilson et al. 2019 Coral Reefs), driving up richness on both protected and fished reefs.

The lines between data points in 1994 and 2005 in Figure S2 (now S3), and Figure S11 have been removed as suggested.

I also had an issue with the conclusion that “Any influence of protection on the reef benthos appears to have been lost,” (p8). Proving that reserves have a positive effect on reefs trajectories is challenging, requiring multiple conditions including a reserve effect on a driving factor (eg herbivore abundance) and a significant period of time between disturbances to document reef trajectories. Thus while reserve effects on coral cover are theoretically obvious, they have rarely been documented. The authors suggest reserve effects on coral cover pre-bleaching, but acknowledge this may be due to placement issues (reserves are often put on the best reefs). The results post-bleaching suggest this is more likely than an actual reserve effect since there is no difference between protected and fished reefs when all reefs are set to low coral cover levels by bleaching. Hence I don’t think it is true that the influence of protection on the benthos has been lost, but rather it wasn’t clear it existed at all on these reefs for whatever reason.

We have edited the text in this paragraph, removing “appears to have been lost”, and to more clearly acknowledge the uncertainty regarding higher pre-bleaching coral cover in reserves. However, we have also noted that one of the reserves was established for bird conservation on the island (in 1968), rather than for the surrounding coral reefs per-se, suggesting that selective placement for high coral cover was not a factor there. The three sites in this reserve also had ~15% higher coral cover than fished sites in the pre-bleaching data.

I think the results need to be better interpreted for what they mean for management – in particular, if someone was setting up new reserves how do these results inform siting decisions? It also seems to me worth pointing out that the regime-shifted results might be particularly relevant in the Caribbean where shifts to macroalgal dominance are typical. Furthermore, it should be pointed out that in theory the effects are only short- to medium-term – presumably, at least on recovering reefs, macroalgal cover will decrease, coral cover and structure will increase and the fish assemblage will shift back toward carnivorous species?

Thank you for this suggestion. We have added two paragraphs to the end of the main text to discuss how science and reef management may need to adapt under uncertain future configurations, drawing on recent review / conceptual papers also thinking in this space (Hughes et al. 2017 Nature, Bellwood et al. 2019 Biol Conserv; Williams & Graham 2019 Func Ecol Special Issue). We have included siting issues as part of this discussion, and the relevance of the regime-shifted results to the Caribbean. Unfortunately, we do not believe the results will be short lived as

severe coral bleaching events are increasing in frequency (Hughes et al. 2018 Science), and so the dynamics we show here continue. We have included some discussion on this also.

Given these results, I don't really see why "existing management strategies need to be re-evaluated, and the potential for new approaches should be investigated." This may be true, but I don't see how it is supported by the results here – reserves still generally have useful, positive benefits?

Agreed – reserves will still have an important role, and be part of the management tool kit. We had not meant to suggest they were no longer useful at all, and think this is now clearer with the new two paragraphs at the end of the main text.

REVIEWERS' COMMENTS:

Reviewer #1 (Remarks to the Author):

After carefully checking on data and codes, reading the reviewed ms and the detailed answers to reviewers comments, I have no more critical issues. The authors did a pretty good job revising it. The manuscript is well analyzed and nicely written, giving important results that add to the literature.

Reviewer #2 (Remarks to the Author):

I have read the rebuttal letter, looked at the changes made, and re-read the paper, and am happy that the authors have addressed all of my concerns.